# A Hermite Surface Triangle Modeling Method Considering High-Precision Fitting of 3D Printing Models

**Ruichao Lian** [1], **Shikai Jing** [2,*] 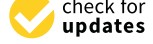, **Yang Chen** [2] and **Jiangxin Fan** [2]

1 Institute of Advanced Structure Technology, Beijing Institute of Technology, Beijing 100081, China; 7520210053@bit.edu.cn
2 School of Mechanical Engineering, Beijing Institute of Technology, Beijing 100081, China; 3120205251@bit.edu.cn (Y.C.); fanjx0321@163.com (J.F.)
* Correspondence: jingshikai@bit.ed; Tel.: +86-13321166219

**Abstract:** Three-dimensional printing is a layer-by-layer stacking process. It can realize complex models that cannot be manufactured by traditional manufacturing technology. The most common model currently used for 3D printing is the STL model. It uses planar triangles to simplify the CAD model. This approach makes it difficult to fit complex surface shapes with high accuracy. The fitting result usually suffers from loss of local features of the model, poor fitting accuracy, or redundant data due to face piece subdivision, which will cause problems such as poor manufacturing accuracy or difficult data processing. To this end, this paper proposes a method for constructing Hermite surface models considering high-precision fitting of 3D printing models. The mapping relationship between different surface triangles and the same base triangle is established by analyzing the characteristics of Hermite surface triangles in AMF format files and using the radial variation property. By constructing a cubic surface model with general parameters and combining the vertex and tangent vector information, a cubic Hermite curve and surface triangle model are obtained. A sampling mapping point solution method is proposed, which transforms the volume integration problem between models into the summation problem of sampling point height difference. Considering the mean deviation and variance in multiple directions of the sampling points, a method for calculating and evaluating the model fitting error is constructed. Finally, the effectiveness of the proposed method is verified by rabbit and turbine.

**Keywords:** 3D printing; Hermite; surface model; high-precision fitting; error

**MSC:** 08C05; 14P99; 51P05



## 1. Introduction

Three-dimensional printing is an advanced manufacturing technology that enables the "free fabrication" of complex structures quickly and efficiently with a simple device [1,2]. Compared with traditional manufacturing processes, it overcomes the limitations of complex configurations that are difficult to machine and reduces processing procedures, manufacturing cycle time, and manufacturing costs [3]. In recent years, 3D printing technology has been successfully applied in aerospace, automotive, and other areas [4–8]. Although 3D printing technology has made breakthroughs, due to its unique manufacturing process, the manufactured products usually have errors problems, which greatly restrict the widespread use of the technology [9].

There are three main sources of errors that exist in the 3D printing manufacturing process. The first error comes from the conversion between model formats, i.e., the process of converting a computer-aided design (CAD) model to a model in the format required for 3D printing; the second error comes from the layered slicing and path planning algorithm [10]; and the third error comes from the manufacturing accuracy of the device itself [11]. All of the above errors directly affect the molding accuracy of the final printed

structure. However, compared to the latter two errors, if the first error cannot be effectively reduced, it will be difficult to manufacture a high-precision structure even if the subsequent process is highly accurate.

When performing 3D printing, we first need to build a digital model through modeling software. However, the model formats generated by different modeling software vary. They are not directly used to drive 3D printers. In order to generate Gcode that can "communicate" directly with the 3D printer, the model needs to be converted from different formats to the common STereoLithography (STL) format file for 3D printing. Currently, 3D printing models usually use planar triangles to form an envelope model to represent the CAD model, such as the more widely used STL model. It approximates the CAD model by setting the maximum chord height between the planar triangle and the surface of the model [12,13]. For this reason, when the CAD model has complex surfaces or high local accuracy, using planar triangles to simplify it will inevitably result in a loss of features and accuracy of the model [14]. In order to retain the features and accuracy of the CAD model as much as possible, the triangular facets of the overall model need to be continuously subdivided during the format conversion process. This will cause problems such as the too-large amount of model data or data redundancy, which inevitably increases the difficulty of model data processing [15]. Compared with the planar triangle model, the surface triangle model has higher degrees of freedom through parametric shape control. It can fit the surface and complex features of the model with high accuracy by using a relatively small number of face pieces, which effectively solves the problems caused by the simplification of the overall model by planar triangles. Several scholars have investigated the construction methods of surface triangles. Vlachos et al. [16] proposed a point-normal triangle in order to improve the visual quality in graphics rendering. Its main idea is to use a Bezier surface triangle (e.g., PN triangle) surface to replace each triangle in the original mesh. Compared with other Bezier triangles, PN triangles have lower degrees of freedom, and their shapes are influenced not only by the normal vectors but also by their different methods. Hamann et al. [17] constructs a $C^0$ continuous surface by using a triangular rational quadratic Bezier surface to approximate a cubic linear interpolation function profile. The construction of Bezier surfaces requires control point information, which is harder to obtain directly when performing model fitting. NURBS and B-sample surfaces using surface approximation control meshes all have a similar problem to Bezier surfaces in that it is difficult to construct a direct mathematical relationship between the surface model and the original model.

Unlike surfaces using control meshes, surface shape control based on boundary conditions is simpler and more intuitive, and easier to achieve stitching between surfaces. Márta et al. [18] proposed a new definition of a surface that uses three triangular surfaces instead of the original boundary curves on the triangular parameter domain to generate a triangular surface. This interpolation scheme has affine transformation invariance [19], while the connection between the resulting surface and its components is continuous along a common boundary curve, except for the vertices. This method involves a tremendous amount of data input and also contains the combined operation of three surfaces, which greatly increases the computational cost. In addition, Hagen [20] proposes an interpolation method based on the Hermite operator, which implements the interpolation of the boundary curvature of an arbitrary triangle.

In order to meet the growing demand for model formats for 3D printing, the American Society for Testing and Materials (ASTM) Special Advisory Panel has creatively proposed a surface triangle in the additive manufacturing file (AMF) format [21,22]. The surface triangle consists of cubic Hermite curves [23], but they only define the boundary curves of the surface triangle and do not define the Hermite surface triangle model completely. According to the authors' knowledge, there are few studies on Hermite surface triangles, but compared with other surface triangles, the definition and input quantity of Hermite surface triangles are relatively easy. It is not only suitable for 3D printing the required multi-surface sheet model, but also can make full use of the surface information contained

in the original design model to achieve high accuracy fitting of the model. Therefore, how to construct a Hermite surface triangle model and make full use of the existing 3D printing manufacturing model information to fit the CAD model with high accuracy is a problem worthy of study.

Based on the above analysis, this paper proposes a Hermite surface triangle model construction method considering the high-precision fitting of 3D printing models. Affine transformation is used to establish the mapping relationship between multiple surface triangles and feature triangles. Then, the cubic surface model with general parameters is constructed and the cubic Hermite surface model is solved using the vertex and tangent vector information. Finally, the model fitting error calculation and comprehensive evaluation are realized by using the height difference between the model sampling points and the mapping points.

## 2. Hermite Surface Characterization

### 2.1. Definition of Hermite Curve in AMF

AMF is a format file that supports 3D printing, which contains the multi-color, multi-material, honeycomb structures and properties, etc. Its structure is similar to the STL file, which is a collection of several small spatial triangular facets. It is a model shell formed by combining triangular facets together after the triangular meshing of a 3D solid model. Each of its triangular face pieces consist of three vertices that obey the right-hand rule and whose corresponding normal vectors are directed outward. According to the AMF standard file, each edge of a surface triangle is a cubic Hermite curve, and the construction of each surface triangle depends on the Hermite curve of the boundary. Each Hermite curve is then determined by the position information of the triangle vertices recorded in the AMF file and the normal or tangential vector information. The surface triangles in the AMF file can be defined by the vertex and endpoint tangent vectors $t_{ij}$ of the edges, as shown in Figure 1a. It can also be defined directly by the normal vectors $n_i$ on the vertices, as shown in Figure 1b.

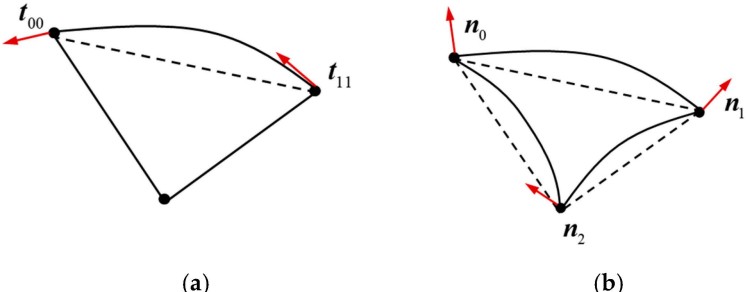

(a)                                                                                     (b)

**Figure 1.** Definition of AMF surface triangles: (**a**) tangent vector of the vertex; (**b**) normal vectors of the vertices.

In the surface triangle definition of the AMF file, the cubic Hermite curve consists of two endpoints and their tangent vectors, as shown in Figure 2.

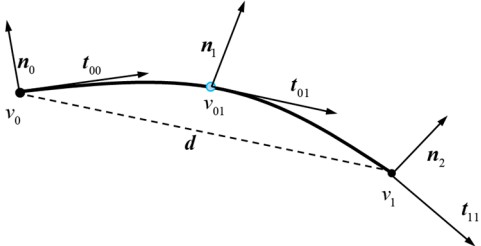

**Figure 2.** Hermite curve.

The mathematical expression of the cubic Hermite curve [24–26] can be described as:

$$h_e(s) = \left(2s^3 - 3s^2 + 1\right)v_0 + \left(s^3 - 2s^2 + s\right)t_0 + \left(-2s^3 + 3s^2\right)v_1 + \left(s^3 - s^2\right)t_1 \quad (1)$$

where $v_0$ and $v_1$ are the two endpoints of the curve. $t_0$ and $t_1$ are the two tangent vectors in the direction of the Hermite surface. The formula for calculating the tangent vector at any point on the curve can be formulated as:

$$t(s) = \left(6s^2 - 6s\right)v_0 + \left(3s^2 - 4s + 1\right)t_0 + \left(-6s^2 + 6s\right)v_1 + \left(3s^2 - 2s\right)t_1 \quad (2)$$

when a normal vector is defined at the endpoint, it can be converted to a tangent vector by Equation (3).

$$t_0 = |d_0| \frac{-(n_0 \times d_0) \times n_0}{|(n_0 \times d_0) \times n_0|} \quad (3)$$

where $d_0 = v_1 - v_0$. $n_0$ and $n_1$ are the normal vectors of the corresponding endpoints, respectively.

### 2.2. Determine Feature Triangle and Mapping Relationship

The Hermite surface triangle is constructed by means of cubic Hermite curves based on the vertices. The triangle is essentially parametric curves and each curve involves a parameter that takes values in the range [0, 1]. Therefore, its shape is theoretically related to the three parameters $\eta$, $\xi$ and $\tau$, and there is a certain correlation between $\eta$, $\xi$ and $\tau$, as shown in Figure 3.

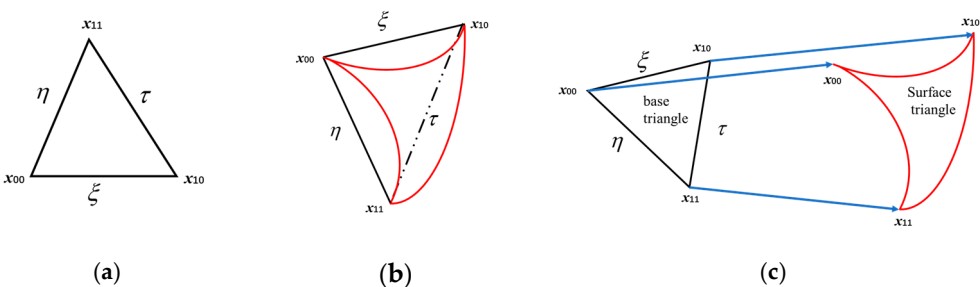

| (a) | (b) | (c) |

**Figure 3.** Curved triangle parameters: (**a**) parameters of each side; (**b**) curved triangle; (**c**) mapping of parameter fields.

The shape of the Hermite surface triangle is influenced by the location of its vertices, the direction of the tangent vectors, and the size of the tangent vectors. Each surface triangle has a different shape, so the shape of the base triangle formed by the three vertices of the surface triangle is also different. This would introduce too many parameters and increase the computational cost. To facilitate the calculation, the affine transformation is used in this subsection to simplify the calculation of the parameters in the surface triangle. Affine transformation is the process of transforming to another vector space by performing one linear transformation (multiplying by one matrix) and one translation (adding one vector) in the vector space. The basic idea of the proposed method is to use affine transformation to establish a mapping relationship between surface triangles, base triangles, and specified feature triangle. It can effectively normalize the complex problem and reduce the computational cost. More knowledge about affine transformations can be found in the literature [27].

According to the affine transformation property, all triangles can be obtained by the affine transformation of feature triangles. As shown in Figure 4, in the right-angle parameter domain, the feature triangle with right-angle characteristics and two right-angle sides of unit length is constructed with (0, 0), (1, 0), and (1, 1) as vertices, which satisfies the

principle of regular shape and simple calculation. It effectively reduces the three parameter variables of the surface triangle to two. The feature triangle can be described as:

$$C = \{(u,v)\,|\,0 \le v \le u \le 1\} \tag{4}$$

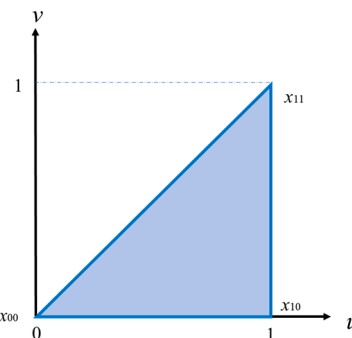

**Figure 4.** Feature triangle.

Based on the affine transformation property, any surface triangle and base triangle have a mapping relationship with the feature triangle $C$, i.e., their three vertices correspond to the three vertices of the base triangle, as shown in Figure 5.

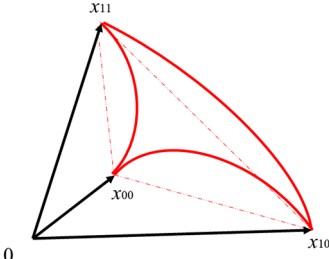

**Figure 5.** Vertex one-to-one correspondence.

The mapping relationship between the surface triangles and the feature triangles is described by $S(u,v)$, as shown in Figure 6. $S(u_i, v_i)$ is the corresponding point of any point $(u_i, v_i)$ on the characteristic triangle in the surface triangle.

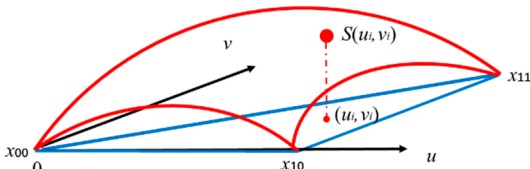

**Figure 6.** Mapping relationship between feature triangle and surface triangle.

The mapping relation is essentially a parametric formulation of surface triangles, as it satisfies the one-to-one correspondence between surface triangles and characteristic triangle vertices. Therefore, the mapping relation between them can be described as:

$$\begin{aligned} x_{00} &= S(0,0) \\ x_{10} &= S(1,0) \\ x_{11} &= S(1,1) \end{aligned} \tag{5}$$

The model is greatly simplified by Equation (5), which can effectively reduce the complexity of subsequent calculations. It is worth noting that for different surface triangles, the corresponding $S(u,v)$ is different.

### 3. Hermite Surface Triangle Model

In this subsection, based on the constructed feature triangles and mapping relations, the general parametric surface model is first established by the two parameters $u$ and $v$. Then, the vertex coordinates and normal vectors of the Hermite curve are matched with the surface model to build the Hermite surface triangle model.

Let $S(u, v)$ be a general two-parameter surface. To ensure the smoothness of the surface, set it as a cubic equation. Then, its mathematical expression can be described as:

$$\begin{aligned} S(u, v) \quad &= C_{00} + C_{10}u + C_{01}v + C_{20}u^2 + C_{11}uv + C_{02}v^2 \\ &+ C_{21}u^2v + C_{12}uv^2 + C_{30}u^3 + C_{03}v^3 \end{aligned} \tag{6}$$

where $C_{ij}$ is the three-dimensional characteristic coefficient.

$$C_{ij} = \begin{bmatrix} C_{ij}^x \\ C_{ij}^y \\ C_{ij}^z \end{bmatrix}, i, j \in [0, 3] \tag{7}$$

As shown in Figure 7, from the vertex coordinates and tangent vector information of the surface triangles, it is known that $v = 0$, $u = 1$ and $u = v$ are the three boundary curve parameters characterized by the vertices $x_{00}$, $x_{10}$, and $x_{11}$ in the counterclockwise direction, respectively. Let $S(u, 0)$, $S(1, v)$, and $S(u, u)$ be their corresponding curves, respectively.

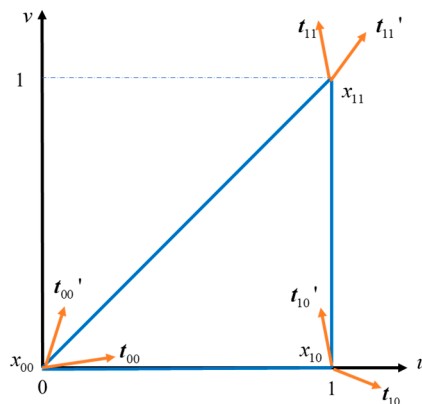

**Figure 7.** Coordinates and tangent vectors of the vertex of a curved triangle.

Substitute the characteristic parameters $v = 0$, $u = 1$, and $u = v$ into Equation (6). Then, the three boundary curves in the general parametric surface can be expressed as:

$$\begin{aligned} S(u, 0) \quad &= C_{00} + C_{10}u + C_{20}u^2 + C_{30}u^3 \\ S(1, v) \quad &= C_{00} + C_{10} + C_{01}v + C_{20} + C_{11}v + C_{02}v^2 \\ &+ C_{21}v + C_{12}v^2 + C_{30} + C_{03}v^3 \\ S(u, u) \quad &= C_{00} + C_{10}u + C_{01}u + C_{20}u^2 + C_{11}u^2 \\ &+ C_{02}u^2 + C_{21}u^3 + C_{12}u^3 + C_{30}u^3 + C_{03}u^3 \end{aligned} \tag{8}$$

The Hermite boundary curve corresponding to Equation (8) is obtained by substituting the vertex and tangent vector information in Figure 8 into Equation (1).

$$\begin{aligned} S(u, 0) &= (2u^3 - 3u^2 + 1)x_{00} + (u^3 - 2u^2 + u)t_{00} + (-2u^3 + 3u^2)x_{10} + (u^3 - u^2)t_{10} \\ S(1, v) &= (2v^3 - 3v^2 + 1)x_{10} + (v^3 - 2v^2 + v)t_{10}' + (-2v^3 + 3v^2)x_{11} + (v^3 - v^2)t_{11} \\ S(u, u) &= (2u^3 - 3u^2 + 1)x_{00} + (u^3 - 2u^2 + u)t_{00}' + (-2u^3 + 3u^2)x_{11} + (u^3 - u^2)t_{11}' \end{aligned} \tag{9}$$

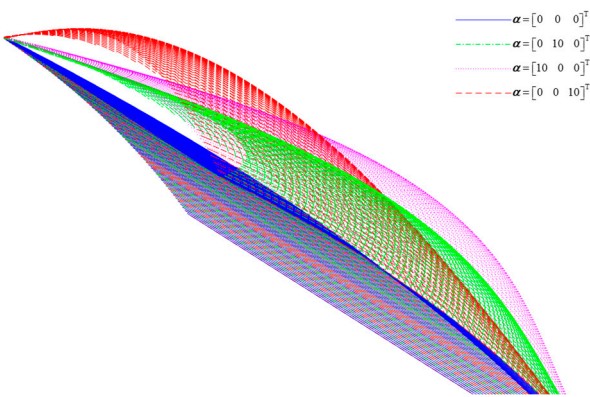

**Figure 8.** Hermite surfaces for different values of $\boldsymbol{a}$.

Equations (8) and (9) are combined to obtain Equation (10).

$$
\begin{cases}
\begin{aligned}
&C_{00} + C_{10}u + C_{20}u^2 + C_{30}u^3 = (2u^3 - 3u^2 + 1)x_{00} + (u^3 - 2u^2 + u)t_{00} \\
&+(-2u^3 + 3u^2)x_{10} + (u^3 - u^2)t_{10} \\
&C_{00} + C_{10} + C_{01}v + C_{20} + C_{11}v + C_{02}v^2 + C_{21}v + C_{12}v^2 + C_{30} + C_{03}v^3 \\
&= (2v^3 - 3v^2 + 1)x_{10} + (v^3 - 2v^2 + v)t_{10}' + (-2v^3 + 3v^2)x_{11} + (v^3 - v^2)t_{11} \\
&C_{00} + C_{10}u + C_{01}u + C_{20}u^2 + C_{11}u^2 + C_{02}u^2 + C_{21}u^3 + C_{12}u^3 + C_{30}u^3 + C_{03}u^3 \\
&= (2u^3 - 3u^2 + 1)x_{00} + (u^3 - 2u^2 + u)t_{00}' + (-2u^3 + 3u^2)x_{11} + (u^3 - u^2)t_{11}'
\end{aligned}
\end{cases}
\tag{10}
$$

The eigencoefficients of the Hermite surface model can be obtained by solving Equation (10).

$$
\begin{aligned}
C_{00} &= x_{00} \\
C_{10} &= t_{00} \\
C_{01} &= t_{00}' - t_{00} \\
C_{20} &= -3x_{00} - 2t_{00} + 3x_{10} - t_{10} \\
C_{11} &= a \\
C_{02} &= -2t_{00}' + 3x_{11} - t_{11}' + 2t_{00} - 3x_{10} + t_{10} - a \\
C_{21} &= t_{10}' - t_{00}' + t_{00} - a \\
C_{12} &= 2t_{00}' + t_{11}' - 2t_{10}' - 2t_{00} - t_{10} - t_{11} + a \\
C_{30} &= 2x_{00} + t_{00} - 2x_{10} + t_{10} \\
C_{03} &= 2x_{10} + t_{10}' - 2x_{11} + t_{11}
\end{aligned}
\tag{11}
$$

Then, Equation (6) can be written as Equation (12).

$$
S(u,v) = S'(u,v) + f_a(u,v)\boldsymbol{a}
\tag{12}
$$

$$
\begin{aligned}
S'(u,v) &= C_{00} + C_{10}u + C_{01}v + C_{20}u^2 + C_{02}'v^2 \\
&+ C_{21}'u^2v + C_{12}'uv^2 + C_{30}u^3 + C_{03}v^3
\end{aligned}
\tag{13}
$$

$$
\begin{aligned}
C_{21}' &= t_{10}' - t_{00}' + t_{00} \\
C_{02}' &= -2t_{00}' + 3x_{11} - t_{11}' + 2t_{00} - 3x_{10} + t_{10} \\
C_{12}' &= 2t_{00}' + t_{11}' - 2t_{10}' - 2t_{00} - t_{10} - t_{11}
\end{aligned}
\tag{14}
$$

where $f_a(u,v) = v(u-v)(1-u)$. $\boldsymbol{a} = [a_x\ a_y\ a_z]^{\mathrm{T}}$ is the set shape vector, which aims to improve the controllability of the surface model, and thus achieve the adjustment of the accuracy of fitting the local details of the CAD model. Figure 8 shows the change in the shape of the surface triangle when $\boldsymbol{a}$ is taken to different values. The focus of this paper is to construct a Hermite surface triangle for 3D printing with high-precision model fitting. A discussion of $\boldsymbol{a}$ will not be developed in this paper, and the specific details will be reflected in another paper. In order to verify the validity of the constructed surface model fit, we set $\boldsymbol{a} = \begin{bmatrix} 0 & 0 & 0 \end{bmatrix}^{\mathrm{T}}$ in the subsequent examples.

## 4. Error Calculation and Evaluation Method

In order to verify the effectiveness of the proposed method, the error calculation and comprehensive evaluation method between the fitted model and the CAD model are presented in this section. The fitting error is essentially the offset that exists between the fitted model and the original

model surface. It is theoretically more accurate to use the inter-model volume error as the evaluation criterion for the offset of the model. However, since the distance $H(x, y)$ between model surfaces is related to both its location $(x, y)$ and surface data, and the original model shape has uncertainty, this will make the fitting error difficult to be uniformly expressed by mathematical formulas. To this end, the fitting error is calculated in this paper by means of model sampling. The main idea is to transform the problem of integration of the volumes between the two models into a problem of summing the height differences at the sampling points.

In order to sample the data points of the original model reasonably and comprehensively, a method of calculating the sampling mapping points is proposed. The sampled data points are obtained by meshing the CAD model using *Hypermesh* software, as is shown Figure 9.

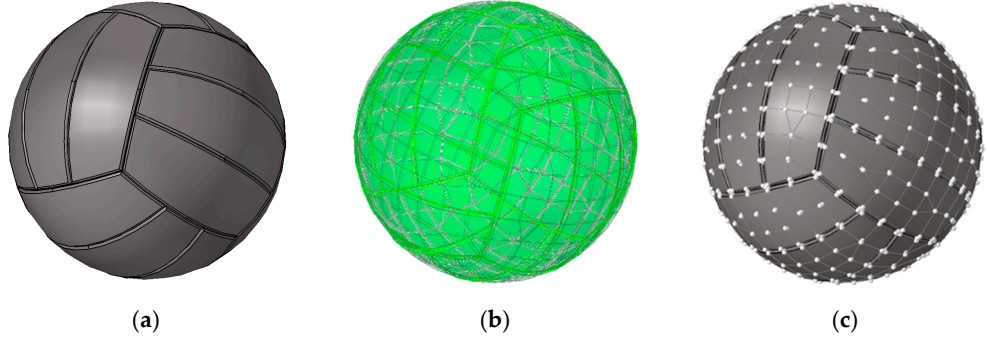

|            (**a**)            |            (**b**)            |            (**c**)            |

**Figure 9.** Sampling point acquisition for basketball model: (**a**) CAD model; (**b**) grid division; (**c**) sampling data points.

Figure 10 shows the schematic diagram of the model fitting error. $P_O(x_O, y_O, z_O)$ is any sampling point in the CAD model, and the fitting error at that point is the height difference between that point and the corresponding mapping point of the fitted model. The mapping point is the intersection point between a vertical line $L_O$ made along the X/Y/Z direction past the $P_O(x_O, y_O, z_O)$ point and the fitted model, as shown in Figure 10. $P_H(x_O, y_O, z_H)$ and $P_S(x_O, y_O, z_S)$ are the mapping points of $P_O(x_O, y_O, z_O)$ on the Hermite surface model $M_H$ and STL model $M_S$, respectively.

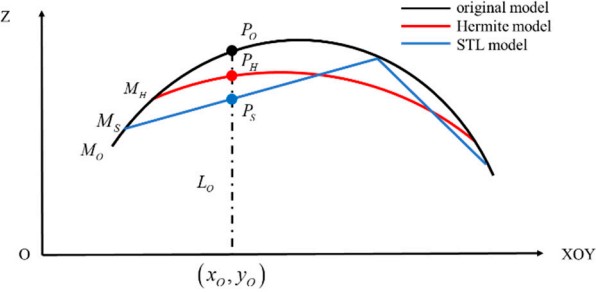

**Figure 10.** Local schematic of model fitting error.

The coordinates of the mapped point can be calculated from the information of the triangular surface slice where the vertical line intersects the fitted model. As shown in Figure 11, $P_O{}'$ is the projection of the sampled points. $\overrightarrow{P_O{}'P_i{}'}$, $\overrightarrow{P_O{}'P_j{}'}$, and $\overrightarrow{P_O{}'P_k{}'}$ are the vectors constructed with the $\Delta P_i{}'P_j{}'P_k{}'$ vertices in a counterclockwise direction starting from $P_O{}'$, respectively. $\overrightarrow{P_O{}'P_i{}'}$, $\overrightarrow{P_O{}'P_j{}'}$, and $\overrightarrow{P_O{}'P_k{}'}$ can be regarded as vectors rotating counterclockwise with $P_O{}'$ as the center. When $P_O{}'$ lies within $\Delta P_i{}'P_j{}'P_k{}'$, $\overrightarrow{P_O{}'P_i{}'} \times \overrightarrow{P_O{}'P_j{}'}$, $\overrightarrow{P_O{}'P_j{}'} \times \overrightarrow{P_O{}'P_k{}'}$, and $\overrightarrow{P_O{}'P_k{}'} \times \overrightarrow{P_O{}'P_i{}'}$ have the same sign, indicating that the vertical line $L_O$ intersects the triangle $\Delta P_iP_jP_k$. Conversely, when $P_O{}'$ is outside $\Delta P_i{}'P_j{}'P_k{}'$, the vertical line $L_O$ does not intersect with triangle $\Delta P_iP_jP_k$.

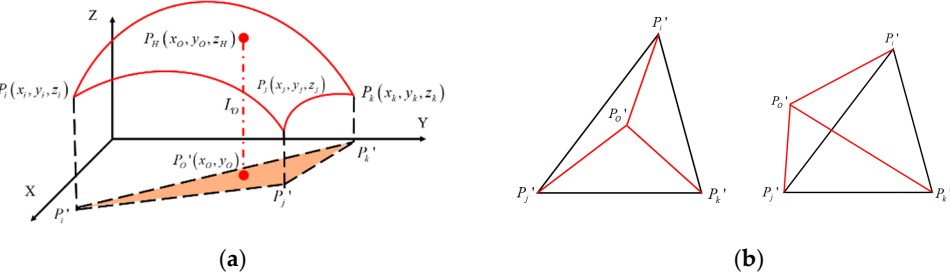

**Figure 11.** The vertical line intersects the curved triangle: (**a**) the projection point (Z-direction); (**b**) location of projection point.

Taking a vertical line along the Z-direction as an example, when the triangle in the Hermite model intersecting the vertical line is determined, the parameter value $(u_H, v_H)$ of the sampled mapping point on the Hermite surface triangle can be found by Equation (16). The vertical coordinates of the sampling mapping points (e.g., Equation (16)) can be obtained by substituting $(u_H, v_H)$ into Equation (14).

$$
\begin{cases}
x_O &= C_{00}^x + C_{10}^x u_H + C_{01}^x v_H + C_{20}^x u_H{}^2 + C_{11}^x u_H v_H \\
&\quad + C_{02}^x v_H{}^2 + C_{21}^x u_H{}^2 v_H + C_{12}^x u_H v_H{}^2 + C_{30}^x u_H{}^3 + C_{03}^x v_H{}^3 \\
y_O &= C_{00}^y + C_{10}^y u_H + C_{01}^y v_H + C_{20}^y u_H{}^2 + C_{11}^y u_H v_H \\
&\quad + C_{02}^y v_H{}^2 + C_{21}^y u_H{}^2 v_H + C_{12}^y u_H v_H{}^2 + C_{30}^y u_H{}^3 + C_{03}^y v_H{}^3
\end{cases}
\tag{15}
$$

$$
\begin{aligned}
z_H &= C_{00}^z + C_{10}^z u_H + C_{01}^z v_H + C_{20}^z u_H{}^2 + C_{11}^z u_H v_H \\
&\quad + C_{02}^z v_H{}^2 + C_{21}^z u_H{}^2 v_H + C_{12}^z u_H v_H{}^2 + C_{30}^z u_H{}^3 + C_{03}^z v_H{}^3
\end{aligned}
\tag{16}
$$

It is worth noting that the procedure calculates all the sampled mapping points on the surface triangles intersecting the vertical line and takes the point closest to the sampled point as the error calculation point.

In the Z-direction of the STL fitted model, the triangles with vertices $P_i(x_i, y_i, z_i)$, $P_j\left(x_j, y_j, z_j\right)$ and $P_k(x_k, y_k, z_k)$, where the sampled mapping point $P_S(x_O, y_O, z_S)$ is located, should satisfy the following conditions.

$$
\begin{vmatrix}
x - x_i & y - y_i & z - z_i \\
x_j - x_i & y_j - y_i & z_j - z_i \\
x_k - x_i & y_k - y_i & z_k - z_i
\end{vmatrix} = 0
\tag{17}
$$

The vertical coordinate of this sample mapping point can be described as:

$$
z_S = \frac{-a(x_O - x_i) - b(y_O - y_i)}{c} + z_i
\tag{18}
$$

where

$$
\begin{aligned}
a &= (y_j - y_i)(z_k - z_i) - (y_k - y_i)(z_j - z_i) \\
b &= (z_j - z_i)(x_k - x_i) - (z_k - z_i)(x_j - x_i) \\
c &= (x_j - x_i)(y_k - y_i) - (x_k - x_i)(y_j - y_i)
\end{aligned}
\tag{19}
$$

After obtaining the sampling mapping points on the fitted model, the error between the fitted model and the original model can be obtained by calculating the distance between the sampling points and the mapping points, and the unit of distance is by millimeter. In order to make a comprehensive evaluation of the error of the fitted model, the evaluation method of calculating the fitting error in terms of mean deviation and variance is proposed. In the Z-direction, the mean deviation (mm) and variance (mm$^2$) of the sampled points can be described as:

$$
\overline{e_Z} = \frac{\left|H_1^Z\right| + \left|H_2^Z\right| + \ldots + \left|H_n^Z\right|}{n}
\tag{20}
$$

$$
s_z{}^2 = \frac{\left(\left|H_1^Z\right| - \overline{e_z}\right)^2 + \left(\left|H_2^Z\right| - \overline{e_z}\right)^2 + \ldots + \left(\left|H_n^Z\right| - \overline{e_z}\right)^2}{n - 1}
\tag{21}
$$

where $H_{ij}^Z(x, y) = M_i(x, y) - M_j(x, y)$. $M_i$ and $M_j$ are the original model and the fitted model, respectively. $H$ is the distance difference between the models at the point $(x, y)$. $n$ is the number

of sampling point. Similarly, the fitting errors in the X and Y directions can be solved by the above method.

## 5. Numerical Cases

**Case 1.** *In this section, we take the Stanford classical rabbit model with high complexity morphology as an example and compare the fitting error of its Hermite surface model and the 3D printing generic STL model. As shown in Figure 12, the solid rabbit model in step format was imported into Hypermesh software, and the sampled data points with 702 located on the model surface were obtained by meshing.*

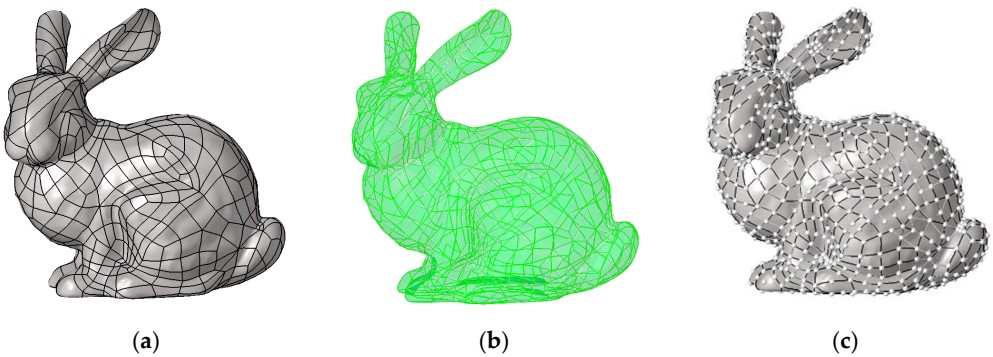

(**a**)                              (**b**)                              (**c**)

**Figure 12.** Sampling point acquisition for rabbit model: (**a**) solid model; (**b**) grid division; (**c**) sampling data points.

The rabbit model was fitted using the Hermite surface triangles constructed in this paper and the planar triangles of the STL, respectively. The number of triangular surface slices is controlled by changing the approximation tolerance and mesh density of the fit. And finally, five groups of Hermite and STL fitted models were obtained, corresponding to the number of triangular facets of 16,862, 18,519, 22,820, 25,323 and 27,239, respectively. Table 1 shows the sampling mean deviation and variance of the original model and the fitted model in the X, Y, and Z directions.

**Table 1.** Results of model fitting errors.

| Number | Models | Number of Facets | Z-Direction | | Y-Direction | | X-Direction | |
|---|---|---|---|---|---|---|---|---|
| | | | Mean Deviation (mm) | Variance (mm²) | Mean Deviation (mm) | Variance (mm²) | Mean Deviation (mm) | Variance (mm²) |
| 1 | Hermite STL | 16,862 | 0.29 13.17 | 4.15 5561 | 0.89 3.47 | 16.20 59.78 | 0.64 2.31 | 25.06 102.10 |
| 2 | Hermite STL | 18,519 | 0.27 22.19 | 3.92 215,450 | 1.77 3.12 | 160.76 53.73 | 0.79 3.50 | 38.81 174.12 |
| 3 | Hermite STL | 22,820 | 0.50 2.06 | 16.88 57.23 | 0.92 3.16 | 23.84 57.51 | 0.43 2.10 | 9.66 112.78 |
| 4 | Hermite STL | 25,323 | 0.26 2.59 | 3.70 306.76 | 0.60 2.64 | 9.13 40.47 | 0.47 1.81 | 10.15 67.52 |
| 5 | Hermite STL | 27,239 | 0.25 2.76 | 3.67 259.87 | 0.84 2.69 | 21.53 45.57 | 0.38 2.84 | 8.80 461.31 |

Theoretically, as the number of fitted model facets increases, the accuracy of the fit should also improve, i.e., the mean deviation will be reduced. However, as shown in Table 1, there is no such rule between the obtained data. The main reason for the existence of this phenomenon is that when the fitted model is not finely drawn to the local features of the original model, the triangle that should be used to calculate the fitting error at its closest distance does not intersect with the vertical line after some of the sampling points make a vertical line along the error calculation direction. This will cause an offset in the mapping point selection, resulting in a large sampling error.

Tables 2 and 3 show the distribution of the sampling mean deviation of the data in groups 1 and 5, respectively. As shown in Table 2, the maximum fitting error of the Hermite surface model in the Z direction is around 50, and then the overall fitting error is mostly distributed below 5. In contrast, the maximum fitting error of the STL model reaches more than 800, and there is still a large distribution around 50. In the other two directions, the maximum fitting errors of the two are similar, but the Hermite surface model has a smaller value of sampling error and a relatively concentrated

distribution. In contrast, the fitting errors of the STL model are relatively discrete in distribution and have larger values. Similarly, as shown in Table 3, the maximum fitting error is similar in the Y direction, but in the Z and X directions, the maximum fitting error of the STL model is much larger than that of the Hermite surface model, and the overall error distribution is more discrete in all three directions.

As shown in Tables 1–3, the overall fitting accuracy of the Hermite surface model is higher compared to the STL model, but the obtained fitting error fluctuates more due to the existence of mapping point bias, which does not truly reflect the deviation between the fitted model and the original model. In order to evaluate the fitting ability of the two models more accurately and objectively, the distance-weighted nearest neighbor algorithm [28] is used to process the data in Table 1. This method achieves the screening and removal of data with more discrete distribution by calculating the distance between objects and assigning larger weights to closer distances. Table 4 and Figure 13 show the fitting errors and the variation in mean deviation of mean deviation after removing the "noise points", respectively.

**Table 2.** Mean deviation in the three directions in group 1.

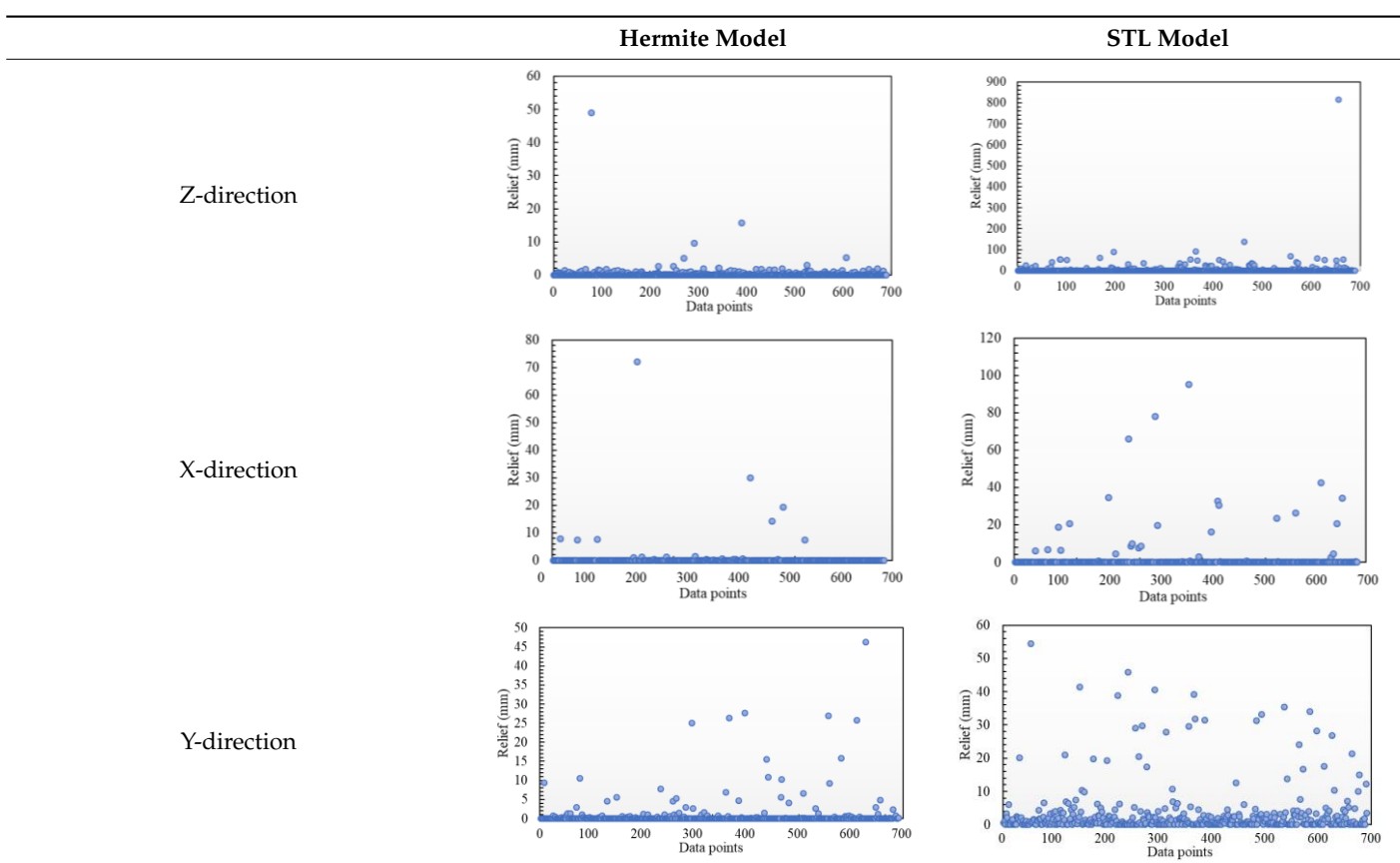

As can be seen from Table 4 and Figure 13, the mean deviation between the Hermite and STL models in the Z, X, and Y directions gradually decrease as the number of model facets increases, and gradually stabilize at the 4th and 5th groups. It shows that the accuracy of the fit of the 2 to the model is gradually improving with the increase in the number of face slices. The fitting error of the STL model fluctuates in the Y-direction, e.g., the error becomes larger in group 3. The main reason for this phenomenon is that STL uses planar triangles for fitting, which is more likely to produce an offset in the mapping points. At the same time, the overall fitting accuracy of the Hermite surface model is much better than that of the STL planar model, which can preserve the original model characteristics and accuracy as much as possible with a smaller number of face slices. This not only can effectively solve the problem of excessive data volume or data redundancy caused by the continuous subdivision of the face slices in the STL model when improving the fitting accuracy, but also can further improve the processing efficiency and manufacturing accuracy of complex models in the 3D printing process.

**Table 3.** Mean deviation in the three directions in group 5.

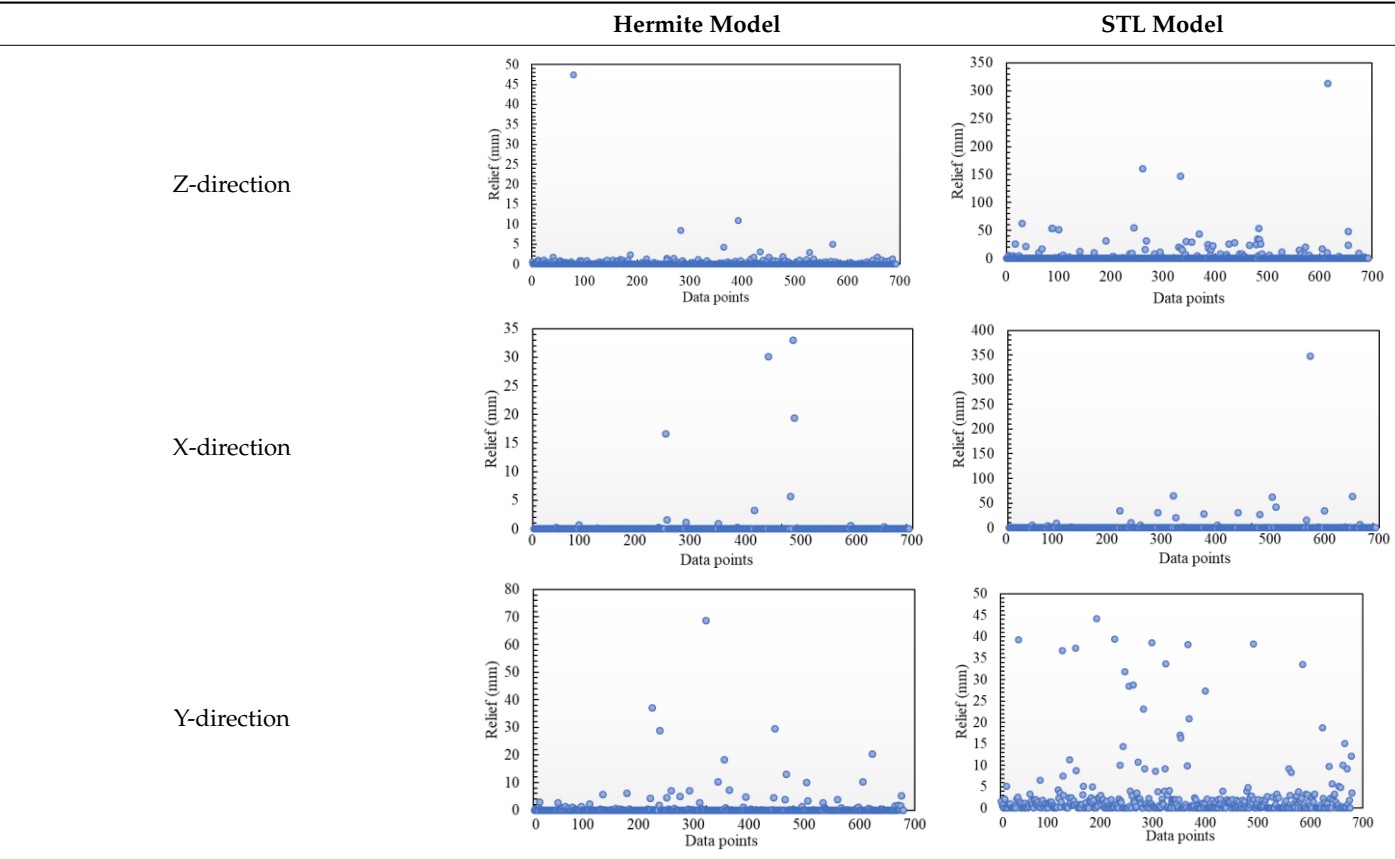

| | Hermite Model | STL Model |
|---|---|---|
| Z-direction | | |
| X-direction | | |
| Y-direction | | |

**Table 4.** Fitting error results after removing the "noise" from the rabbit model.

| Number | Models | Number of Facets | Z-Direction | | Y-Direction | | X-Direction | |
|---|---|---|---|---|---|---|---|---|
| | | | Mean Deviation (mm) | Variance (mm²) | Mean Deviation (mm) | Variance (mm²) | Mean Deviation (mm) | Variance (mm²) |
| 1 | Hermite | 16,862 | 0.18 | 0.23 | 0.32 | 1.43 | 0.15 | 0.91 |
| | STL | | 0.40 | 2.33 | 1.39 | 3.21 | 0.28 | 1.80 |
| 2 | Hermite | 18,519 | 0.16 | 0.22 | 0.33 | 1.32 | 0.13 | 0.74 |
| | STL | | 0.38 | 2.15 | 1.20 | 2.86 | 0.27 | 1.87 |
| 3 | Hermite | 22,820 | 0.10 | 0.49 | 0.23 | 0.71 | 0.02 | 0.04 |
| | STL | | 0.33 | 2.01 | 1.44 | 4.04 | 0.21 | 1.32 |
| 4 | Hermite | 25,323 | 0.09 | 0.31 | 0.24 | 0.65 | 0.02 | 0.17 |
| | STL | | 0.29 | 1.74 | 1.32 | 3.81 | 0.17 | 1.09 |
| 5 | Hermite | 27,239 | 0.08 | 0.29 | 0.28 | 1.07 | 0.03 | 0.05 |
| | STL | | 0.29 | 1.73 | 1.20 | 3.22 | 0.16 | 0.84 |

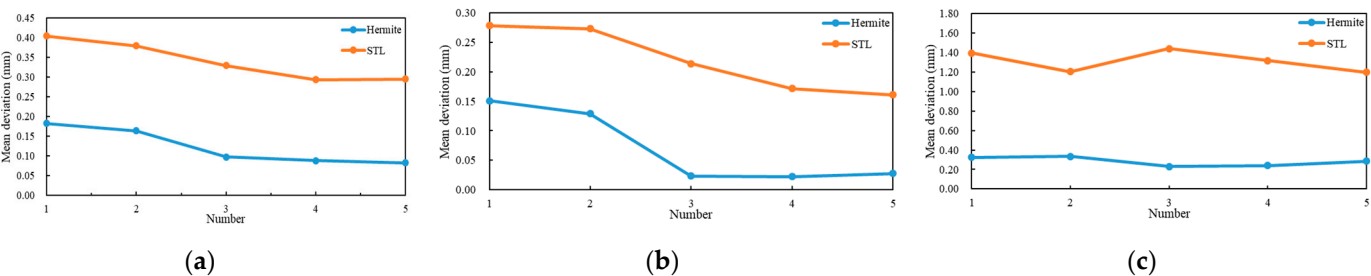

(**a**)          (**b**)          (**c**)

**Figure 13.** Variation in mean deviation: (**a**) Z-direction; (**b**) X-direction; (**c**) Y-direction.

**Case 2.** *The turbine model is used as another example to verify the effectiveness of the proposed method. A Hermite surface model and a 3D printing generic STL model are used to fit it, and the fitting error is analyzed. The solid model of the turbine and the 950 sampled data points extracted based on the grid division are shown in Figure 14.*

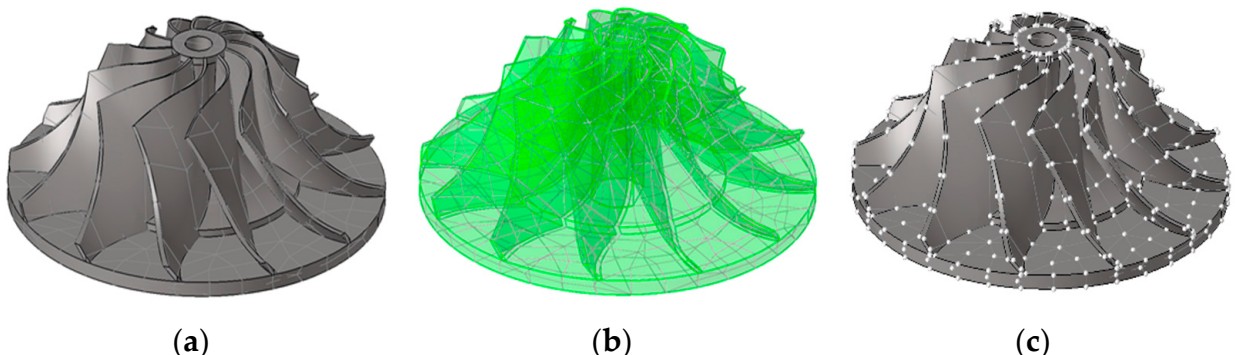

(a)      (b)      (c)

**Figure 14.** Sampling point acquisition for turbine model: (**a**) CAD model; (**b**) grid division; (**c**) sampling data points.

**Table 5.** Fitting error results after removing the "noise" from the turbine model.

| Number | Models | Number of Facets | Z-Direction | | X-Direction | | Y-Direction | |
|---|---|---|---|---|---|---|---|---|
| | | | Mean Deviation (mm) | Variance (mm$^2$) | Mean Deviation (mm) | Variance (mm$^2$) | Mean Deviation (mm) | Variance (mm$^2$) |
| 1 | Hermit | 18,272 | 0.25 | 0.08 | 0.18 | 0.07 | 0.24 | 0.07 |
| | STL | | 0.35 | 0.13 | 0.30 | 0.19 | 0.33 | 0.16 |
| 2 | Hermit | 20,540 | 0.26 | 0.08 | 0.12 | 0.02 | 0.25 | 0.07 |
| | STL | | 0.38 | 0.13 | 0.24 | 0.12 | 0.36 | 0.13 |
| 3 | Hermit | 22,752 | 0.23 | 0.05 | 0.13 | 0.06 | 0.24 | 0.07 |
| | STL | | 0.37 | 0.14 | 0.26 | 0.18 | 0.33 | 0.16 |
| 4 | Hermit | 24,336 | 0.20 | 0.07 | 0.12 | 0.08 | 0.23 | 0.08 |
| | STL | | 0.35 | 0.12 | 0.25 | 0.12 | 0.33 | 0.19 |
| 5 | Hermit | 28,592 | 0.21 | 0.07 | 0.11 | 0.10 | 0.20 | 0.04 |
| | STL | | 0.34 | 0.12 | 0.25 | 0.12 | 0.32 | 0.08 |

The Hermite and STL fitting errors were compared by five groups of models with face piece numbers of 18,272, 20,540, 22,752, 24,336, and 28,592, respectively. Table 5 shows the mean deviation and variance of the fitted model in three directions after removing the "noise". The variation of the mean deviation is shown in Figure 15.

As is shown in Table 4 and Figure 13, the fitting error is gradually decreasing as the number of model facets increases in the Z and Y directions. The model fit error was minimized in group 5. The mean deviation was about 0.2 and 0.35 for Hermite and STL, respectively. In the X-direction, the fitting error of Hermite ranged from 0.2 to 0.1, and gradually tended to 0.1 as the number of model facets increased. The fitting error of the STL model showed a gradual decrease with the increase in the number of model facets, except for a slight fluctuation due to the mapping point offset on group 3. However, its lowest error among the 5 groups is still around 0.25. The overall fitting accuracy of the Hermite model for this case is still better than that of the STL model, which verifies the effectiveness of the proposed method.

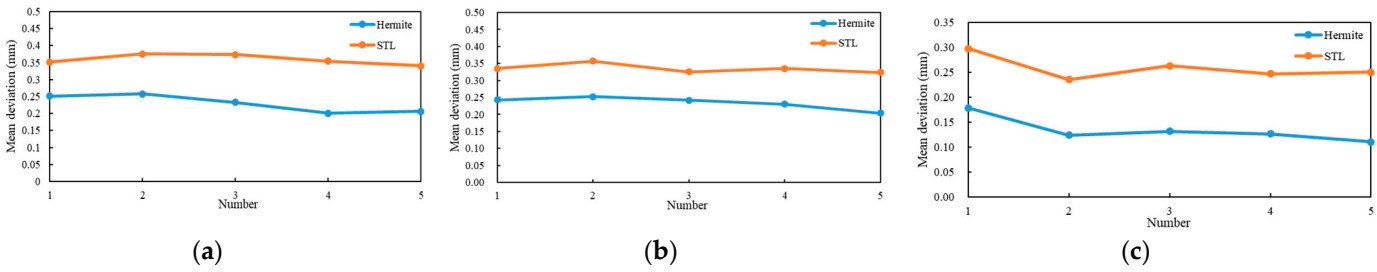

(a)      (b)      (c)

**Figure 15.** Variation in mean deviation: (**a**) Z-direction; (**b**) X-direction; (**c**) Y-direction.

## 6. Conclusions

In this paper, we propose a Hermite surface triangle model construction method considering the high-precision fitting of 3D printing models. The mapping relationship between different surface triangles and characteristic triangles is established by radial variation. Using the vertex and tangent vector information, a cubic Hermite curve model with adjustable accuracy of the model local fitting is constructed based on the general parametric cubic surface model. The model effectively reduces the parameter variables, simplifies the complexity of the calculation, and achieves the specification of the solution problem. A model fitting error calculation and evaluation method based on sampling mapping points is proposed. It transforms the continuous integration into a discrete summation problem, effectively solving the problem that is difficult to express uniformly in mathematical formulas due to the uncertainty of the original model. The effectiveness of the proposed method in improving the model fitting accuracy was verified by using rabbit and turbine models with five different sets of face slices. In the future research work, we will carry out a systematic study of *a*. Considering the different characteristics of the model, we will study an adaptive control method of *a* that can satisfy high-precision fitting. In addition, based on the constructed Hermite surface model, we will study the adaptive layering technique of the surface model and surface path planning method considering the layered slicing and path planning errors. This will provide vital support to further enhance efficient and high-precision manufacturing of 3D printing.

**Author Contributions:** Conceptualization, R.L.; formal analysis, R.L.; investigation, J.F.; methodology, J.F.; writing—original draft, R.L. and J.F.; writing—review and editing, S.J., Y.C. and R.L. All authors have read and agreed to the published version of the manuscript.

**Funding:** The work described in this paper was jointly funded by the National Key Research and Development Program of China under Grant No. 2017YFB1102804.

**Data Availability Statement:** Not applicable.

**Conflicts of Interest:** The authors declare no conflict of interest.

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
