# Peer review of "A Hermite Surface Triangle Modeling Method Considering High-Precision Fitting of 3D Printing Models"

_axioms, doi:10.3390/axioms12040370_

Round 1

Reviewer 1 Report

Dear Authors

 This is an interesting article about the modeling method considering high precision fitting of 3D printed models.

The mentioned problem is important to the issue of improving the accuracy of 3D printed parts.

However, the article has serious flaws and improvement of the methodology and presentation of results is needed.

 The main question is:

Why not use the CAD model to directly control the 3d printing process?

Why “converting a CAD model to a model in the format required for 3D printing” is really needed?

Major issues:

There is a lack of references about the second and third error sources.

 Presentation of results:

What are the units of presented values in Tables 1-3 ? (mm, inches,..)

 The results presented in Table 1 are incoherent. There are some strong outliers in the data.

 “In order to evaluate the fitting ability of the two models more accurately and objectively, we removed some "noise points" (e.g., data with a more discrete distribution) that obviously affected the error in the data in Table 1.”

The arbitrary removing of outstanding data is not objective, and thus methodologically incorrect!

Other comments:

Line 51

“For this reason, when the CAD model has complex surfaces or high local accuracy, using planar triangles to simplify it will inevitably result in a loss of features and accuracy of the model” … which is particularly noticeable in the case of printing sharp edges.

 There is a contradiction between the descriptions in line 80 “… an interpolation method based on the Hermite operator, which implements the interpolation of the boundary curvature of an arbitrary triangle. The above method involves a tremendous amount of data input and also contains the combined operation of three surfaces, which greatly increases the computational cost”, and line 91 “compared with other surface triangles, the definition and input quantity of  Hermite surface triangles are relatively easy”

 Minor issues:

 Not described abbreviations: CAD, STL, 

illegible symbols on the Figure 8

Figure 13 and 15 are hardly readable.

There is a new version of the standard [19]

Comments: The direction of further research should be mentioned including the parameter a.

A spell check of English language and style is required (e.g. Normoal )

Reviewer 2 Report

Dear,

Kind regards!

Reviewer 3 Report

Dear Authors

I am sending comments attached.

Kind regards

Reviewer

Round 2

Reviewer 1 Report

Dear Authors

Thank you for the detailed response of the review comments.

I know the answers for the first two questions, but it may be interesting for the readers and should be included into paper.

According to point 5 about deleting some noise points-it raises some doubts, therefore, additional descriptions of the procedure should be included (Is it performed manually or procedurally?)

Additional question:

All deviation in results have positives values. Can there be negative deviations?

There still are illegible symbols (Chinese?) on the Figure 8
